# The Key Role of Hepcidin-25 in Anemia in Multiple Myeloma Patients with Renal Impairment

**DOI:** 10.3390/medicina58030417

**Published:** 2022-03-11

**Authors:** Małgorzata Banaszkiewicz, Jolanta Małyszko, Krzysztof Batko, Ewa Koc-Żórawska, Marcin Żórawski, Paulina Dumnicka, Artur Jurczyszyn, Karolina Woziwodzka, Aleksandra Maleszka, Marcin Krzanowski, Andrzej Kraśniak, Ryszard Drożdż, Katarzyna Krzanowska

**Affiliations:** 1Department of Nephrology and Transplantology, Jagiellonian University Medical College, 30-688 Kraków, Poland; mbanaszkiewicz92@gmail.com (M.B.); batko.krzysztof@gmail.com (K.B.); woziwodzka.karolina@gmail.com (K.W.); mkrzanowski@op.pl (M.K.); akrasnia@cm-uj.krakow.pl (A.K.); 2Department of Nephrology, Dialysis and Internal Medicine, Medical University of Warsaw, 02-091 Warsaw, Poland; jolmal@poczta.onet.pl; 3Second Department of Nephrology and Hypertension with Dialysis Unit, Medical University of Bialystok, 15-276 Bialystok, Poland; ewakoczorawska@wp.pl; 4Department of Clinical Medicine, Medical University of Bialystok, 15-254 Bialystok, Poland; mzorawski@wp.pl; 5Department of Medical Diagnostics, Jagiellonian University Medical College, 30-688 Kraków, Poland; paulina.dumnicka@uj.edu.pl (P.D.); ryszard.drozdz@uj.edu.pl (R.D.); 6Department of Hematology, Jagiellonian University Medical College, 31-501 Kraków, Poland; mmjurczy@cyf-kr.edu.pl; 7Department of Diagnostics, University Hospital in Kraków, 30-688 Kraków, Poland; amaleszka@su.krakow.pl

**Keywords:** multiple myeloma, soluble transferrin receptor, anemia, hepcidin 25, renal impairment, tumor microenvironment, biomarker

## Abstract

*Background and objectives:* Anemia is common in multiple myeloma (MM) and is caused by a complex pathomechanism, including impaired iron homeostasis. Our aim is to evaluate the biomarkers of iron turnover: serum soluble transferrin receptor (sTfR) and hepcidin-25 in patients at various stages of MM in relation with markers of anemia, iron status, inflammation, renal impairment and burden of the disease and as predictors of mortality. *Materials and methods:* Seventy-three MM patients (six with smoldering and 67 with symptomatic disease) were recruited and observed for up to 27 months. Control group included 21 healthy individuals. Serum sTfR and hepcidin were measured with immunoenzymatic assays. *Results:* MM patients with and without anemia had higher sTFR compared to controls, while only anemic patients had higher hepcidin-25. Both hepcidin-25 and sTfR were higher in anemic than non-anemic patients. Higher hepcidin-25 (but not sTfR) was associated with increasing MM advancement (from smoldering to International Staging System stage III disease) and with poor response to MM treatment, which was accompanied by lower blood hemoglobin and increased anisocytosis. Neither serum hepcidin-25 nor sTfR were correlated with markers of renal impairment. Hepcidin-25 predicted blood hemoglobin in MM patients independently of other predictors, including markers of renal impairment, inflammation and MM burden. Moreover, both blood hemoglobin and serum hepcidin-25 were independently associated with patients’ 2-year survival. *Conclusions:* Our results suggest that hepcidin-25 is involved in anemia in MM and its concentrations are not affected by kidney impairment. Moreover, serum hepcidin-25 may be an early predictor of survival in this disease, independent of hemoglobin concentration. It should be further evaluated whether including hepcidin improves the early diagnosis of anemia in MM.

## 1. Introduction

Multiple myeloma (MM) is a plasma cell disorder that accounts for approximately 10% of hematological malignancies. It is a more common disease in the elderly population and often develops following an asymptomatic stage. The diagnostic process involves the search for a myeloma-defining event, including anemia and renal impairment. However, recent guidelines [1] allow for earlier diagnosis of MM preceding the development of organ damage in patients with high (≥60%) clonal plasma cell involvement of bone marrow, involved to uninvolved serum free light chains (FLCs) ratio ≥100 and serum FLCs ≥100 mg/L, or more than one focal lesion in magnetic resonance imaging. Median overall survival of patients with newly diagnosed disease reaches 6 to 8 years, which is a striking improvement attributed to modern therapy [2]. The focus on timely diagnosis and prevention of organ damage drives the interest in diagnostic and prognostic biomarkers. In theory, a validated biomarker can act as a surrogate of underlying disease pathways and inform the physician on the alterations in organ-specific processes; for example, our previous studies demonstrated that urinary insulin growth factor binding protein 7 (IGFBP-7), neutrophil gelatinase-associated lipocalin (NGAL) monomer, and transgelin-2 may be markers of renal impairment in patients with MM [3,4]. The use of appropriate biomarkers may be crucial for adequate tailoring of the treatment plan. However, in order to be widely adopted into practice, novel biomarkers need to offer a significant improvement as compared to the current ones.

Anemia is commonly observed in MM (in about 70% of patients), and is even more prevalent in those patients who develop renal impairment (almost 90% of such cases). Anemia may be caused by complex pathomechanisms. Cytokines produced by plasmacytes lead to anemia of chronic disease (ACD), by erythropoiesis inhibition and impaired iron homeostasis [5]. The most common laboratory findings in MM-related anemia and ACD are: (1) normocytic and normochromic anemia, (2) normal to mildly low serum iron levels, (3) high serum ferritin, and (4) hemosiderin in bone marrow macrophages [6]. It seems that inadequate excretion of erythropoietin (EPO) compared to the degree of anemia, reduction of erythrocytes’ survival time (<10%), inadequacy of ferric management, and direct suppression of erythropoiesis by neoplastic cells are mainly responsible for the development of ACD [7]. Other MM-related factors leading to anemia include: displacement of erythroid system by neoplastic plasmacytes, proinflammatory activity of cytokines, and disabled apoptosis of the erythroid system. However, renal failure should also be taken into consideration as an important factor contributing to anemia. Among MM patients with renal impairment, lower serum EPO levels appear more frequently (up to 60%) than in patients with normal renal function [8]. The molecules engaged in iron metabolism, namely hepcidin and soluble transferrin receptor (sTfR), have been previously suggested as potential biomarkers in MM, enabling to better characterize the underlying etiology of anemia [9].

Hepcidin, as a one of acute phase proteins, is regulated by interleukin-6 (IL-6), a cytokine inducing MM development (a potential growth factor for myeloma cells). Hepcidin-25 (an active hormone consisting of 25 aminoacids) controls iron delivery from intestinal cells to blood and regulates iron transport and its release from macrophages. There is a potential association between novel drugs in MM and this marker [9]. Moreover, hepcidin expression is regulated by growth differentiation factor 15 (GDF-15), secreted by tumor microenvironment cells. In our previous study, increased GDF-15 was associated with end-organ damage and adverse prognosis in MM patients [7].

sTfR prevents the organism from iron-associated toxicity. TfR1 form has higher affinity for transferrin and is overexpressed on cells with a high rate of proliferation, including malignant hematopoietic cells. Interestingly, sTfR to ferritin ratio was decreased in advanced stages of hematopoietic malignancies [10,11]. Studies have demonstrated increased sTfR serum levels, decreased ferritin and increased sTfR to ferritin ratio in correlation with increasing stages of chronic kidney disease (CKD) [12].

The primary aim of this study was to evaluate biomarkers of iron turnover, sTfR, and hepcidin-25 in patients at various stages of MM progression, and examine their relationship with indicators of anemia and iron status (hemoglobin, red blood cell indices, ferritin, and serum iron concentration) and renal impairment (serum creatinine, estimated glomerular filtration rate–eGFR, and urinary NGAL). Moreover, we examined the relationships of these two markers with indicators of inflammation (IL-6) and burden of this disease (FLCs, β2-microglobulin, and GDF-15). Finally, serum concentrations of sTfR and hepcidin-25 were assessed to verify them as predictors of mortality in multiple myeloma.

## 2. Materials and Methods

### 2.1. Study Design and Patients

The recruitment process and laboratory methodology follow our earlier work [5].

Briefly, in this prospective observational study, patients were recruited during ambulatory control visits at the Departments of Hematology of the University Hospital in Kraków. We included adult patients with smoldering MM (SMM) or MM diagnosis consistent with the International Myeloma Working Group criteria [1]. We excluded patients reporting recent active infection, chronic infection with hepatitis B, C or human immunodeficiency viruses, or diagnosis of active malignancy other than MM and patients on dialysis.

At the start of the study, physicians who conducted ambulatory control visits collected clinical data regarding demographics (patient’s age and sex), history of hematological disease including the date of initial diagnosis of SMM or MM, the current diagnosis, the presence of bone lesions on X-ray, and the information about past and present treatment including the response to the treatment: complete response (CR), partial response (PR), stable disease (SD), or progressive disease (PD), and relevant co-morbidities. Follow-up data on mortality was collected in February 2020, 27 months after the start of the study.

We also recruited a control group consisting of 21 healthy individuals (13 women, 8 men, aged 39–66 years) who provided blood samples used to obtain control results of non-standard laboratory tests.

### 2.2. Ethics Statement

The study was compliant with the Declaration of Helsinki and the International Conference on Harmonization/Good Clinical Practice regulations. Approval by the Bioethics Committee of the Jagiellonian University was granted (approval number 1072.6120.248.2017). Written informed consent was obtained from all participants. All the patients with MM were treated at the Department of Hematology, University Hospital in Kraków, Poland.

### 2.3. Blood Samples and Laboratory Tests

Blood and urine samples were obtained from MM patients at study enrolment, in morning hours, after overnight fast. Blood samples were collected by trained nurses into appropriate closed-system tubes. Urine was self-collected by patients using clean-catch method and dedicated containers.

Routine laboratory tests were performed on the day of blood and urine collection and included complete blood count, serum concentrations of creatinine, iron, total protein, albumin, β2-microglobulin, free light chains, ferritin, serum activity of lactate dehydrogenase, and urine concentrations of light chains.

Sysmex XE 2100 analyzer (Sysmex, Kobe, Japan) was used for complete blood counts. The routine biochemical tests were carried out using automatic biochemical analyzers: Hitachi 917 (Hitachi, Japan) and Modular P (Roche Diagnostics, Mannheim, Germany). The eGFR was calculated based on serum creatinine using the Chronic Kidney Disease–Epidemiology Collaboration (CKD-EPI) 2009 formula. Concentrations of serum FLC, urine LC (κ and λ type) and β2-microglobulin were measured by immunonephelometric method on a BN II analyzer (Siemens GmbH, Erlangen, Germany). Freelite reagents (Binding Site, Birmingham, UK) were used to assess FLCs κ and λ (reference ranges were 1.7–3.7 g/L and 0.9–2.1 g/L, respectively). The immunophenotype of the monoclonal protein was determined by serum immunofixation on agarose gel (EasyFix G26, Interlab, Rome, Italy).

Additional samples for non-routine laboratory tests were obtained along with routine samples’ collection from patients with MM, and were collected using the same methodology from the control subjects. Serum and urine samples for non-routine laboratory tests were aliquoted, frozen, and stored in −70 °C until analysis. Hemolyzed or lipemic serum samples were not used. The non-routine tests were performed in series, with commercially available immunoenzymatic assays (ELISA), and included serum concentrations of sTfR, hepcidin 25, GDF-15, IL-6, and N-terminal prohormone of brain natriuretic peptide (NT-proBNP), and urine concentrations of NGAL monomer.

Serum sTfR was measured using Quantikine IVD ELISA Human sTfR Immunoassay (R&D Systems, Inc., Minneapolis, MN, USA), with the minimum detection range of 0.5 nmol/L. Serum hepcidin-25 was measured using Hepcidin 25 human Cet. No. S-1337 kit (Peninsula Laboratories International, Inc., San Carlos, CA, USA). The detection range for hepcidin 25 was 0.02–25 ng/mL. Serum GDF-15 was measured using Quantikine ELISA Human GDF-15 Immunoassay (R&D Systems, Inc., Minneapolis, MN, USA), with minimum detection dose ≤4.4 pg/mL. Serum IL-6 was measured using Quantikine ELISA Human IL-6 Immunoassay (R&D Systems, Inc., Minneapolis, USA with the minimum detection dose of 0.70 pg/mL. Serum NT-pBNP concentrations were measured by Enzyme-linked Immunosorbent Assay Kit For NT-ProBNP Human (Cloud-Clone Corporation, Huston, TX, USA), with the minimum detection dose of 11.7 pg/mL. Urine NGAL monomer was assessed using Human NGAL monomer-specific ELISA Kit (BioPorto Diagnostics A/S, Hellerup, Denmark), with the minimum detection dose of 10 pg/mL.

### 2.4. Statistical Methods

For categorical data, we reported numbers and percentages of patients. Mean ± standard deviation or median (lower-upper quartile) were reported for normally and non-normally distributed quantitative variables, respectively. The Shapiro–Wilk test was used to assess normality. The differences between groups were assessed with the *t*-test or Mann–Whitney test, according to distribution. Spearman rank correlation coefficient was used to analyze associations between studied quantitative variables and the ordered variable describing MM stage ranging from SMM to ISS stage III. Simple correlations between two quantitative variables were assessed using the Pearson coefficient. We used multiple linear regression to search for independent predictors of blood hemoglobin, including the independent variables that correlated significantly with hemoglobin in the simple analysis. Right-skewed variables were log-transformed using normal logarithm before the correlation and linear regression analyses. Survival times were calculated from the date of the study enrolment until the date of death from any cause or the date of the last follow-up and were estimated with use of the Kaplan–Meier method. Simple and multiple Cox proportional hazard regression were used to study the predictors of overall mortality. The statistical tests were two-tailed and *p* < 0.05 indicated statistical significance. Statistica 12.0 (StatSoft, Tulsa, OK, USA) software was used for computations.

## 3. Results

### 3.1. Baseline Characteristics of the Patients with MM and Differences between Patients with or without Anemia

Seventy-three patients (35 females), aged between 29 and 90 years old, were recruited (Figure 1). Six patients (8%) were diagnosed with smoldering myeloma (SMM) and 67 patients (92%) with symptomatic MM. Forty patients (55% of the studied group) were classified to have stage I disease according to International Staging System for multiple myeloma (ISS) [13]. Eight patients (11%, including the subjects diagnosed with SMM) were treatment naïve upon presentation. Most patients (*n* = 65; 89%) had received at least one line of treatment and 23 (32%) had a history of prior autologous peripheral blood stem cell transplantation. Fifty-two patients (71%) were in complete or partial remission. Fourteen patients (19%) had anemia defined as blood hemoglobin below sex-related lower reference limit (i.e., <11 g/dL in women and <12 g/dL in men). Patients with anemia more commonly received maintenance treatment and demonstrated a worse response to the treatment, as compared to subjects with normal blood hemoglobin (Table 1), while other baseline clinical characteristics were similar across the groups.

Anemia was associated with a higher average mean red blood cell volume (MCV) and anisocytosis (higher red cell distribution width, RDW-CV) (Table 2). Moreover, anemic patients had lower serum albumin, higher β2-microglobulin, and higher FLCs (Table 2). Although serum creatinine and urinary NGAL were elevated in anemic subjects, average eGFR values were similar. Both serum ferritin and soluble sTfR concentrations were higher in MM patients with anemia, resulting in no difference in sTfR/log(ferritin) ratios between anemic and non-anemic subjects (Table 2). There was no difference between the groups in serum iron levels, either. Average serum hepcidin-25 level was more than twice higher in patients with anemia than in those without anemia; similar differences were observed between the groups in serum concentrations of IL-6 and GDF-15 (Table 2).

In the studied 73 patients with MM, serum sTfR was higher than in 21 healthy controls (median (lower; upper quartile)): 25.0 (20.4; 28.4) versus 19.4 (17.3; 23.7) nmol/L, respectively; *p* = 0.002). Both patients with (27.7 (24.6; 33.1); *p* < 0.001)) and without anemia (24.0 (19.9; 27.9); *p* = 0.007) had higher sTfR as compared to controls. Serum concentrations of hepcidin-25 did not differ between MM patients and control subjects: 28.8 (16.5; 44.6) versus 27.1 (20.0; 37.3) ng/mL; *p* = 0.9); however, anemic MM patients (54.2 (30.9; 90.2)) had significantly higher hepcidin-25 than controls (*p* = 0.006).

### 3.2. Associations between the Studied Markers of Iron Metabolism and Baseline Clinical Characteristics of Patients with MM

In the studied patients, no significant association was observed between the ordered variable describing the advancement of MM (from SMM to ISS III) and sTfR concentrations (Figure 2A), while a weak positive association was observed for serum hepcidin-25 (Figure 2B). Increasing severity form SMM to ISS stage III MM was associated with decreasing blood hemoglobin (Figure 2C) and increasing red blood cell anisocytosis (i.e., increasing RDW-CV values; Figure 2D). A weak negative association was observed between serum iron concentrations and the MM stage (R = −0.31; *p* = 0.013). However, there were no associations between the MM stage and other erythrocyte indices (MCV or mean cell hemoglobin, MCH). In addition, we observed no significant associations between MM stage and serum ferritin (R = 0.23; *p* = 0.1) or sTfR/log(ferritin) ratio (R = −0.04; *p* = 0.8).

Serum hepcidin-25 and ferritin concentrations were significantly higher in patients who received maintenance treatment at the start of the study, while the difference was not significant in case of sTfR (Figure 3A–C). Treatment with bortezomib (*p* = 0.019), lenalidomide (*p* = 0.012), and steroid (*p* = 0.008) was associated with significantly higher serum hepcidin-25; treatment with lenalidomide (*p* = 0.004) and steroid (*p* = 0.010) was associated with higher ferritin. This was paralleled by lower hemoglobin (11.71 ± 1.78 versus 13.13 ± 1.61 g/dL; *p* = 0.001) and higher RDW-CV (15.1 (14.3; 17.2) versus 13.9 (13.5; 14.6) %; *p* < 0.001) in treated versus untreated patients. Both serum sTfR and hepcidin-25 concentrations were associated with the response to treatment (Figure 3D,E), whereas other studied markers of iron status did not differ between patients with CR, PR, SD, or PD. Again, patients with CR had the highest blood hemoglobin (*p* = 0.004) and lowest RDW-CV (*p* < 0.001).

Neither of the studied markers of iron metabolism (sTfR, hepcidin-25, ferritin, sTfR/log(ferritin), iron) was associated with sex (*p* > 0.3 in all comparisons) or age (*p* > 0.3 in all cases).

### 3.3. Association between Studied Markers and Blood Hemoglobin in Patients with Multiple Myeloma

In the 73 studied MM patients, blood hemoglobin concentrations were positively associated with serum albumin and eGFR, and negatively associated with β2-microglobulin, serum creatinine, urine NGAL, serum hepcidin-25, GDF-15, and interleukin 6 (Table 3). Neither sTfR (R = −0.14; *p* = 0.2), nor ferritin (R = −0.23; *p* = 0.1), or sTfR/log(ferritin) (R = 0.04; *p* = 0.8) significantly predicted hemoglobin concentrations in the studied group. In a multiple analysis adjusted for sex, response to treatment (CR, PR, SD, or PD) and treatment status (maintenance treatment or no treatment at the start of the study), serum albumin and hepcidin-25 were identified as independent predictors of hemoglobin level (Table 3).

Hepcidin-25 (log-transformed) demonstrated a significant correlation with red blood cell indices: MCV (R = 0.23; *p* = 0.048), MCH (R = 0.24; *p* = 0.040), and RDW-CV (R = 0.33; *p* = 0.004). Moreover, RDW-CV correlated positively with log(ferritin) (R = 0.36; *p* = 0.010), log(interleukin 6) (R = 0.34; *p* = 0.003) and log(GDF-15) (R = 0.42; *p* < 0.001). Iron concentrations (R = 0.31; *p* = 0.012) and log(sTfR) (R = −0.24; *p* = 0.039) were correlated with MCHC.

### 3.4. Relationship between the Studied Markers of Iron Metabolism and their Associations with the Markers of MM Burden, Inflammation and Renal Impairment

Serum hepcidin-25 correlated positively with ferritin (R = 0.61; *p* < 0.001 for log-transformed variables) and, consequently, with sTfR/log(ferritin) ratio (R = −0.38; *p* = 0.006). Otherwise, the studied markers of iron metabolism were not interrelated: serum sTfR did not correlate with ferritin, hepcidin-25, or iron concentrations.

While serum iron concentrations were correlated with renal function (serum creatinine, eGFR, and urine NGAL), we observed no such correlations in the case of serum ferritin, sTfR, and hepcidin (Table 4). Only sTfR and iron concentration significantly correlated with GDF-15 (Table 4). Moreover, serum sTfR negatively correlated with albumin and serum iron positively correlated with β2-microglobulin. No correlations were observed between the studied markers of iron metabolism and IL-6, serum FLC, or urinary light chains. The ratio of sTfR/log(ferritin) did not correlate with any of the studied variables.

### 3.5. Predictors of Mortality in Multiple Myeloma

After 27 months from the start of the study, we collected the data on all-cause mortality. The median observation time in the studied group was 20 months (range: 1 to 25 months; lower; upper quartile: 16; 23 months). Fifteen (21%) patients died during the observation period for the following causes: MM in six, infection in five, other neoplasm in three, and unknown cause in one. The estimated two-year survival rate (Kaplan–Meier method) was 79%. Blood hemoglobin was significantly associated with survival (hazard ratio: 0.67; 95% confidence interval: 0.52–0.87; *p* = 0.003). Baseline serum hepcidin significantly predicted survival in simple analysis (hazard ratio: 1.02; 95% confidence interval: 1.0002–1.04; *p* = 0.047), although the association was weak. Moreover, baseline serum hepcidin-25 appeared to be a negative predictor of survival, independent of hemoglobin concentration (Table 5). No association with survival was observed for serum sTfR (*p* = 0.8).

## 4. Discussion

In this study, we evaluated the relationships between the advancement of MM and biomarkers of anemia and iron turnover. Anemia is a common manifestation of MM. In a retrospective analysis of over one thousand MM patients published by Kyle et al., 73% of patients presented with anemia at diagnosis [14]. Data collected at cancer care centers indicate that the vast majority (85%) of patients experience anemia at some point in time, while its presence is associated with a worse quality of life. In MM patients, hemoglobin levels are usually lower than in other hematological conditions [15]. Developing reliable biomarkers for anemia and clarifying the underlying pathomechanisms represent areas of high interest. Furthermore, differentiating iron deficiency from ACD is useful for adequate treatment tailoring (e.g., potential benefits from iron supplementation and use of erythropoiesis-stimulating agents). Hematopoietic scores based on a composite of platelets <150 × 10^9^/L, hemoglobin <10 g/dL, and high MCV (MCV > 96) represent a simple tool to reliably predict survival in incident cases of MM [16].

In our study, we observed that iron-regulating cytokines and hormones (hepcidin-25, IL-6, and GDF-15) were significantly elevated in anemic subjects with MM. More advanced disease was negatively associated with iron levels and positively with serum hepcidin, while no significant associations were noted with ferritin, sTfR, or sTfR/log(ferritin). Although previously postulated as a useful measure, sTfR/log(ferritin) was of no benefit in our MM patients, as both ferritin and sTfR values were elevated. Iron transport in plasma occurs when transferrin interacts with its membrane receptors, which are prevalent in bone marrow tissue (80%) and are stimulated by iron deprivation. The concentration of the soluble form of the receptor (sTfR) in plasma is proportional to cellular mass and remains stable over time in healthy subjects. No age- or sex-related variations are observed. sTfR can be described as a marker of erythropoiesis when iron stores are adequate and available, whereas in iron deficiency it is a measure of iron status [17]. Hypoferremia with lower availability of iron for erythrocyte progenitors is a potential mechanism underlying anemia in MM [18]. The inflammatory component of malignancy involves cytokine signaling, which, in turn, promotes hepcidin secretion by hepatocytes, ferroportin binding and prevention of iron efflux from enterocytes and reticuloendothelial cells. This process shapes iron restriction in inflammation [9,10,17,18,19,20]. Our findings are consistent with these hypotheses.

The etiology of MM-related anemia is multifactorial. Rapid proliferation of MM cells, tumor microenvironment, and expansion of myeloma in bone marrow are other factors that may contribute to the “iron block” (e.g., by increasing iron demand and limiting the functional erythropoietic compartment) [10,16]. Myeloma cells can promote erythroblast apoptosis [21,22], while cytokines, such as IL-6, impair erythroid maturation and hemoglobin production outside of “iron restriction” pathways [23]. Recent data analyses support the model hypothesis of erythroid apoptosis induction by myeloma cytokines and erythroblastic island destruction by myeloma cells [24].

Progression of anemia in MM has been associated with worsening erythroid hypoplasia, indirectly marked by transferrin receptor levels. Blunted erythropoietin release has also been observed in more severe anemia, and kidney function appears to be an important factor here [25,26]. However, deficient erythropoietin production may occur without kidney impairment. The proposed mechanisms resemble the pathways of anemia of inflammation and CKD [25]. In the present study, in the majority of patients, anemia can be categorized as normocytic with iron replete status and notable anisocytosis. The concurrent elevation of IL-6 and hepcidin-25 levels suggests that elevated sTfR reflects an increased iron demand, which is restricted to the erythroid progenitors through enhanced activity of the IL-6-hepcidin axis [27]. This falls in line with the current model of the anemia of inflammation in MM.

Moreover, one of the most landmark drugs in MM treatment is the proteasome inhibitor, bortezomib, due to its various anti-myeloma effects, such as the induction of apoptosis, alteration of the bone marrow microenvironment, and inhibition of nuclear factor kappa B (NFκB). It seems that it also affects additional cell survival pathways and exerts inhibitory effects on cytokines, such as IL-6 or tumor necrosis factor α (TNFα) [28]. Hepcidin up-regulation has been linked to IL-6 signaling in MM (although IL-6 independent mechanisms are also recognized) and its levels are elevated in both serum and urine samples [20,29]. IL-6 is a crucial player in the inflammatory microenvironment of MM. It is also itself a biomarker of prognostic significance [30,31]. Hepcidin mRNA in monocytes of MM patients is elevated and in untreated cases it correlates negatively with hemoglobin, and positively with ferritin and IL-6 levels [32]. Data from murine models suggest that competition for iron within the bone marrow microenvironment occurs in the initial stages of MM-related anemia, which would make hepcidin- or cytokine-targeted therapy less effective in newly diagnosed patients [33]. This may also influence conclusions from studies based only on incident MM cases. This association could aid in choosing appropriate treatment in the future, especially in relapsing MM, including antibodies targeting myeloma cells’ antigens, nuclear export inhibitors, or histone deacetylase inhibitors [34].

In our study, hepcidin and albumin were independent predictors of hemoglobin levels in multiple analysis adjusted for other variables correlated with hemoglobin, treatment characteristics, and sex. While hypoalbuminemia is a well-established prognostic indicator in myeloma that is associated with disease severity [35], the potential of hepcidin-25 as a predictor of hemoglobin independent of treatment status is a novel finding. Moreover, we observed a significant relationship between hemoglobin and overall survival, with serum hepcidin-25 being the only independent negative predictor of survival among the studied indices of iron status. However, it is necessary to highlight that the observed association between serum hepcidin concentrations and survival must be considered preliminary due to the limitations of our study, and should be verified in larger studies. Hepcidin levels are increased in many malignancies, including MM, which provides an essential source of iron for the survival of neoplastic cells. Malignant cells require increasing cellular iron import by actions of hepcidin on ferroportin downregulation. In MM, hepcidin expression is related to bone morphogenetic protein 2 (BMP2) and IL-6. However, BMP2 is a stronger inducer of hepcidin than IL-6 [36]. Moreover, local levels of hepcidin are lower than serum levels. All of that suggest that most hepcidin in MM derives from the liver [20] It explains our observation of no correlation between hepcidin and IL-6.

High concentrations of hepcidin are observed in both CKD and in MM [29,37]. In CKD, patients with functional iron deficiency have higher hepcidin levels, which are predicted by the degree of inflammation (e.g., levels of ferritin, fibrinogen, IL-6) [37]. However, although in our study serum iron concentrations were correlated with renal function (serum creatinine, eGFR, and urine NGAL), we observed no such correlations in the case of serum ferritin, sTfR, and hepcidin. It is well known that renal impairment in MM reflects poor prognosis and determines the response to chemotherapy [3]. The lack of hepcidin correlation with renal function supports the role of hepcidin as a marker of poor prognosis in MM that is independent of renal injury.

GDF-15 is another potential mediator that plays a role within the myeloma microenvironment. It is secreted by abnormal bone marrow stromal cells [38,39]. The prognostic value of this biomarker has been previously described. GDF-15 concentrations are elevated in advanced MM and decline (alongside with hepcidin levels) following treatment [9,38,39,40]. In the present study, serum hepcidin-25 was associated with red blood cell parameters. We demonstrated that hemoglobin levels correlate not only with hepcidin-25, GDF-15, and IL-6, but also with kidney glomerular (i.e., creatinine as a surrogate) and tubular (i.e., NGAL as a surrogate) function. NGAL is a sensitive marker of kidney function in MM. It is often described as a “real-time indicator of renal injury”. It also shows a relationship with inflammation and tumor burden [3,41,42,43,44].

In the present study, several markers of myeloma burden, kidney injury, inflammation, and iron regulation have been tied to alterations in hemoglobin levels. These observations may reflect a complex network of pathological processes underlying MM. However, dedicated mechanistic study is necessary to evaluate whether these correlations reflect the interaction between molecular pathways or only parallel the progression of disease (and are therefore interrelated). Other limitations are also present and relate to the interpretation of findings. This investigation examined a heterogeneous patient sample at different stages of the disease, with varying degrees of organ involvement and treated with different modalities. These factors may confound our observations. Our conclusions are also based on biomarker assays performed at a single moment in time. The lack of prospective evaluation is a natural limitation of a cross-sectional design, though it is particularly relevant for concentrations of circulating proteins and small molecules, which physiologically fluctuate and can be transiently elevated.

## 5. Conclusions

Multiple myeloma is a progressive hematological malignancy with a poor prognosis and frequent organ impairment. Our results suggest that hepcidin-25 is involved in anemia in MM and its concentrations are not affected by kidney impairment. We postulate that in future, hepcidin-25 may be helpful to early diagnose anemia in MM patients independently of kidney impairment, which in consequence may support early treatment, improving disease management and increasing the quality of life. Moreover, serum hepcidin-25 may be an early predictor of survival in MM, independent of hemoglobin concentration; however, this result must be considered preliminary and needs confirmation in larger studies. In our study, neither sTfR, ferritin, nor sTfR/log(ferritin) were good predictors of hemoglobin concentration in patients with MM. However, MM progression is also driven by other factors, which indicates that a reliable model for disease assessment will require a panel of validated markers.

## Figures and Tables

**Figure 1 medicina-58-00417-f001:**
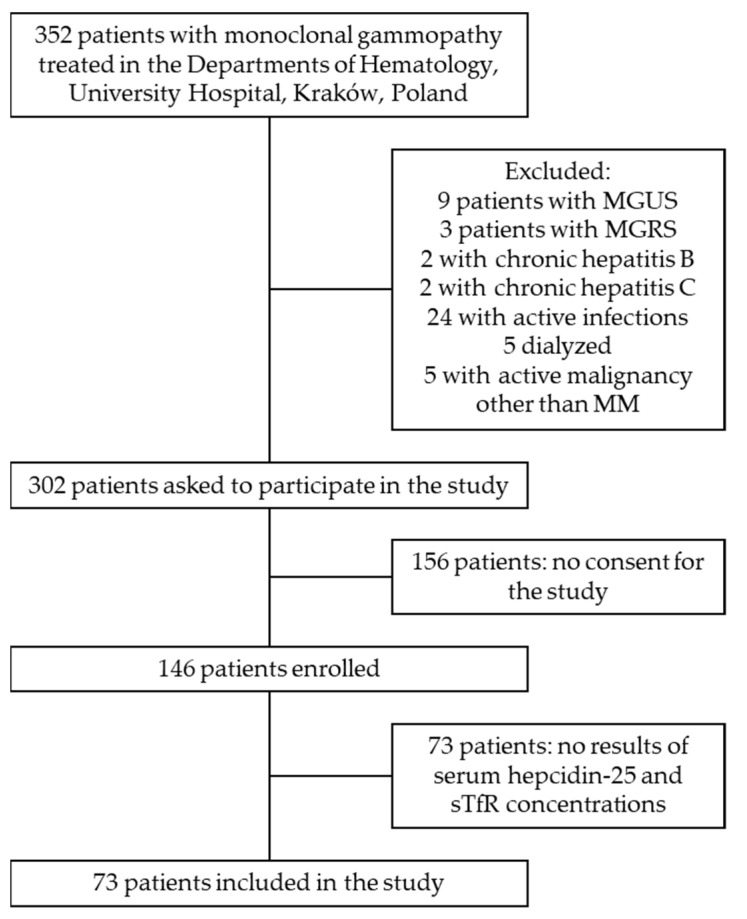
Flow diagram of study patient selection.

**Figure 2 medicina-58-00417-f002:**
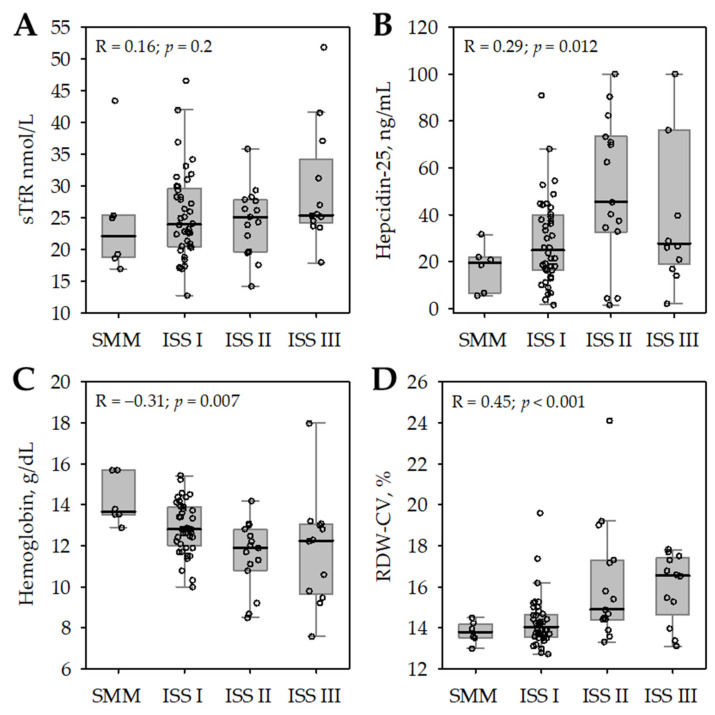
Soluble transferrin receptor (sTfR, **A**), hepcidin-25 (**B**), blood hemoglobin (**C**), and red cell distribution width–coefficient of variation (RDW-CV, (**D**)), among 73 studied patients with multiple myeloma (MM) according to increasing stage of the disease: from smoldering multiple myeloma (SMM) to International Staging System for MM (ISS) stage III. The data are shown as median (central line), interquartile range (box), non-outlier range (whiskers), and outliers (points). Spearman correlation coefficients (R) and *p*-values describe the correlations of the studied variables (shown on the *Y*-axis) with ordered scale of increasing severity of MM (from SMM to ISS stage III MM, as shown on the *X*-axis).

**Figure 3 medicina-58-00417-f003:**
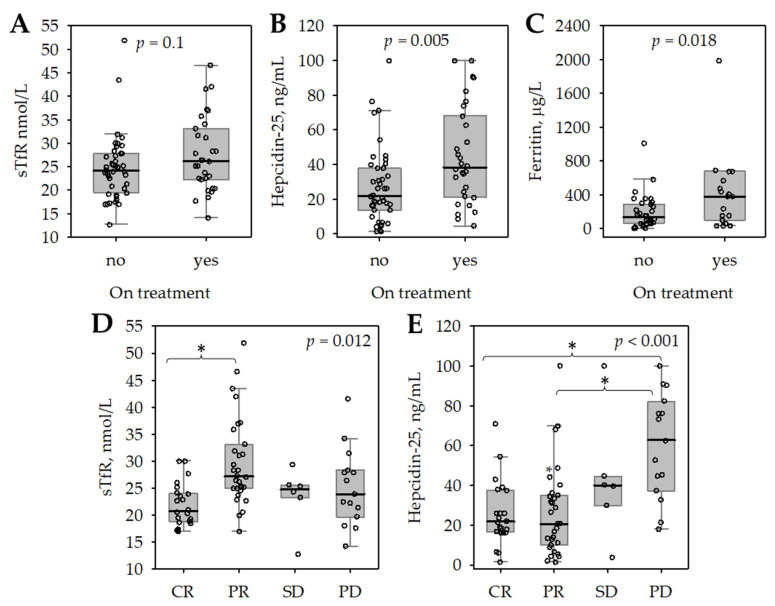
Soluble transferrin receptor (sTfR, (**A**)), hepcidin-25 (**B**), and ferritin (**C**) in serum of studied patients with MM according to treatment status; serum sTfR (**D**) and hepcidin-25 (**E**) in patients with MM according to the response to treatment. The data are shown as median (central line), interquartile range (box), non-outlier range (whiskers), and outliers (points); *p*-values in Mann–Whitney (**A**–**C**) or Kruskal–Wallis (**D**,**E**) test are shown, asterisk depicts significant differences in post-hoc comparisons.

**Table 1 medicina-58-00417-t001:** Baseline demographic and clinical characteristics of studied patients with multiple myeloma (MM) with and without anemia.

Characteristic	MM Patients with Anemia (*n* = 14)	MM Patients without Anemia (*n* = 59)	*p*-Value
Mean age ± standard deviation, years	67.8 ± 10.5	69.0 ± 10.2	0.7
Male sex, *n* (%)	10 (71)	28 (47)	0.1
Median time since diagnosis of MM (lower; upper quartile), months	30 (16; 88)	38 (17; 65)	0.9
Smoldering myeloma, *n* (%)	0	6 (10)	0.07
ISS stage I, *n* (%)	5 (36)	35 (59)	
ISS stage II, *n* (%)	4 (29)	11 (19)	
ISS stage III, *n* (%)	5 (36)	7 (12)	
Immunophenotype:			
IgG, *n* (%)	10 (71)	42 (71)	0.9
IgA, *n* (%)	5 (36)	12 (20)	0.2
IgM, *n* (%)	0	1 (2)	0.9
biclonal, *n* (%)	2 (14)	0	0.9
Free light chains only, *n* (%)	0	1 (2)	0.9
Non-secretory, *n* (%)	0	3 (5)	0.9
Disease state on the day of study visit:			
CR, *n* (%)	0	22 (37)	0.041
PR, *n* (%)	8 (57)	22 (37)	
SD, *n* (%)	1 (7)	5 (8)	
PD, *n* (%)	5 (36)	10 (17)	
Chemotherapy on the day of study visit:			
On maintenance treatment, *n* (%)	10 (71)	20 (34)	0.010
No treatment, *n* (%)	4 (29)	39 (66)	
Treatment with:			
Bortezomib, *n* (%)	3 (21)	7 (12)	0.3
Lenalidomide, *n* (%)	6 (43)	7 (12)	0.006
Thalidomide, *n* (%)	1 (7)	4 (7)	0.9
Cyclophosphamide, *n* (%)	1 (7)	3 (5)	0.8
Melphalan, *n* (%)	0	1 (2)	1.0
Steroid, *n* (%)	10 (71)	19 (32)	0.007
Number of prior treatment schemes:			
No treatment, *n* (%)	0	8 (14)	0.4
1, *n* (%)	3 (21)	14 (24)	
2, *n* (%)	6 (43)	16 (27)	
3 and more, *n* (%)	5 (36)	21 (36)	
History of auto-PBSCT, *n* (%)	6 (43)	17 (29)	0.3
Chronic kidney impairment (eGFR <60 mL/min/1.73 m^2^)	6 (43)	17 (29)	0.3

Abbreviations: CR, complete response; eGFR, estimated glomerular filtration rate; Ig, immunoglobulin; ISS, International Staging System for multiple myeloma; *n*, number of patients; PBSCT, peripheral blood stem cell transplant; PD, progressive disease; PR, partial response; SD, stable disease.

**Table 2 medicina-58-00417-t002:** Baseline results of laboratory tests among studied MM patients with and without anemia. Data are shown as median (lower; upper quartile) or mean ± standard deviation.

Characteristic	MM Patients with Anemia (*n* = 14)	MM Patients without Anemia (*n* = 59)	*p*-Value
White blood cell count, ×10^3^/µL	5.64 (4.35; 6.49)	6.12 (4.94; 7.12)	0.3
Red blood cell count, ×10^6^/µL	3.10 ± 0.38	4.27 ± 0.47	<0.001
Hemoglobin, g/dL	9.90 ± 1.22	13.20 ± 1.27	<0.001
Hematocrit, %	29.3 ± 3.63	38.4 ± 3.53	<0.001
MCV, fL	94.8 ± 5.0	90.2 ± 4.8	0.002
MCH, pg	32.0 ± 1.9	31.1 ± 2.0	0.1
MCHC, g/dL	33.8 ± 0.9	34.3 ± 1.1	0.1
RDW-CV, %	15.5 (14.7; 17.3)	14.2 (13.5; 15.0)	0.001
Platelet count, ×10^3^/µL	162 ± 73	180 ± 60	0.3
Lactate dehydrogenase, U/L	342 (308; 389)	356 (306; 403)	0.7
Albumin, g/L	38.4 (35.7; 41.0)	43.7 (40.6; 45.5)	<0.001
Involved serum free light chains, mg/L	92.7 (27.9; 133.0)	29.9 (16.3; 94.3)	0.047
Involved urine light chains, mg/L	10.09 (6.80; 27.60)	7.83 (6.34; 48.20)	0.9
β2-microglobulin, mg/L	4.93 (2.88; 8.14)	2.53 (2.10; 3.66)	0.006
Serum creatinine, µmol/L	100 (84; 218)	85 (74; 98)	0.025
eGFR (CKD-EPI_Cr_), mL/min/1.73 m^2^	66 (25; 78)	67 (53; 78)	0.3
Urine NGAL, ng/mL	27.7 (11.6; 80.4)	11.2 (4.55; 26.8)	0.019
Iron, µmol/L *	14.4 ± 3.7	15.6 ± 5.2	0.4
Ferritin, µg/L *	380 (284; 568)	136 (55; 350)	0.009
sTfR, nmol/L	27.7 (24.6; 33.1)	24.0 (19.9; 27.9)	0.041
sTfR/log(ferritin) *	4.98 (4.66; 5.22)	4.72 (3.97; 6.98)	0.6
Hepcidin-25, ng/mL	54.2 (30.9; 90.2)	21.9 (13.7; 40.4)	0.002
GDF-15, pg/mL	2450 (1896; 4028)	1141 (802; 1681)	<0.001
Interleukin 6, pg/mL	5.11 (3.28; 7.45)	2.43 (1.53; 4.78)	0.027
NT-proBNP, pg/mL	293.6 (31.2; 460.1)	69.7 (31.6; 199.2)	0.08

* Iron concentrations were available in 63 patients, including 12 with anemia and 51 without anemia. Ferritin concentrations were available in 51 patients, including 9 with anemia and 42 without anemia. Abbreviations: CKD-EPI_Cr_, Chronic Kidney Disease–Epidemiology Collaboration equation based on serum creatinine; eGFR, estimated glomerular filtration rate; GDF-15, growth differentiation factor 15; MCH, mean cell hemoglobin; MCHC, mean cell hemoglobin concentration; MCV, mean cell volume; MM, multiple myeloma; NGAL, neutrophil–gelatinase-associated lipocalin; NT-proBNP, N-terminal pro-B-type natriuretic peptide; RDW-CV, red cell distribution width–coefficient of variation.

**Table 3 medicina-58-00417-t003:** Significant predictors of blood hemoglobin concentrations among studied patients in MM in simple analysis and the results of multiple regression. Multiple regression model was adjusted for sex, treatment status, and the response to treatment.

Independent Variable	Simple Correlation		Multiple Linear Regression	
	R	*p*-Value	β ± SE	*p*-Value
Albumin	0.57	<0.001	0.36 ± 0.12	0.004
β2-microglobulin	−0.38	<0.001	0.03 ± 0.16	0.8
Creatinine	−0.31	0.008	not included	
eGFR	0.33	0.005	−0.02 ± 0.17	0.9
log(NGAL)	−0.30	0.010	−0.02 ± 0.1	0.9
log(hepcidin-25)	−0.39	0.001	−0.27 ± 0.11	0.016
log(GDF-15)	−0.50	<0.001	−0.23 ± 0.17	0.2
log(interleukin 6)	−0.29	0.013	0.02 ± 0.11	0.8
The regression model	Not applicable		R^2^ = 0.49	<0.001

Abbreviations: R, Pearson’s correlation coefficient; β, standardized regression coefficient; SE, standard error; R^2^, coefficient of determination, estimated glomerular filtration rate; GDF-15, growth differentiation factor 15; NGAL, neutrophil-gelatinase associated lipocalin.

**Table 4 medicina-58-00417-t004:** Simple correlations of studied laboratory markers of iron metabolism with markers of renal impairment and tumor burden.

Variable	Log(sTfR)	Log(Hepcidin-25)	Log(Ferritin)	Iron
Creatinine	R = 0.13; *p* = 0.3	R = 0.17; *p* = 0.1	R = 0.16; *p* = 0.3	R = −0.26; *p* = 0.042
eGFR	R = −0.17; *p* = 0.1	R = 0.01; *p* = 0.9	R = 0.05; *p* = 0.7	R = 0.32; *p* = 0.008
log(NGAL)	R = 0.06; *p* = 0.6	R = 0.12; *p* = 0.3	R = 0.06; *p* = 0.7	R = −0.33; *p* = 0.009
Albumin	R = −0.25; *p* = 0.031	R = -0.14; *p* = 0.2	R = -0.16; *p* = 0.2	R = 0.04; *p* = 0.8
β2-microglobulin	R = 0.17; *p* = 0.2	R = 0.18; *p* = 0.1	R = 0.19; *p* = 0.2	R = −0.29; *p* = 0.022
log(GDF-15)	R = 0.27; *p* = 0.019	R = 0.20; *p* = 0.09	R = 0.20; *p* = 0.2	R = -0.32; *p* = 0.010

Abbreviations: eGFR, estimated glomerular filtration rate; GDF-15, growth differentiation factor 15; NGAL, neutrophil–gelatinase-associated lipocalin.

**Table 5 medicina-58-00417-t005:** Simple and multiple Cox proportional hazard regression model to predict two-year overall survival of studied patients with MM.

Independent Variable	Simple Regression		Multiple Regression	
	HR (95% CI)	*p*-Value	HR (95% CI)	*p*-Value
Hemoglobin	0.67 (0.52–0.87)	0.003	0.74 (0.55–0.99)	0.044
Hepcidin-25	1.03 (1.01–1.04)	0.002	1.02 (1.001–1.04)	0.041

Abbreviations: HR, hazard ratio, CI, confidence interval.

## Data Availability

The data is available from the corresponding author upon reasonable request.

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
