# Peer review of "The Key Role of Hepcidin-25 in Anemia in Multiple Myeloma Patients with Renal Impairment"

_medicina, 2022, doi:10.3390/medicina58030417_

Round 1

Reviewer 1 Report

I have read an interesting paper entitled “The key role of hepcidin-25 in anemia in multiple myeloma patients”. It is a well-designed and conducted study, showing authors previous experince in this field. Here are my comments:

  • The authors should try rewriting the Introduction section in order to avoid using multiple myeloma or MM so many time because, in the end, it can be annoying. It is aplicable to the Abstract section , as well.
  • The primary and secondary objectives of this study should be better exposed.
  • The duration of the study should pe presented as a date, not only mentioning that it was about 27 months, in section “study design and patients”
  • A flowchart showing the included and excluded patients can increase the value of the manuscript
  • Lines 334-335: “”from among the studied iron…” – please rewrite
  • Line 375 – “lone myeloma cells…” – what do you mean by that?

Author Response

Manuscript ID: 1618296

 Manuscript title: The Key Role of Hepcidin-25 in Anemia in Multiple Myeloma Patients with Renal Impairment

Response to Reviewer 1

I have read an interesting paper entitled “The key role of hepcidin-25 in anemia in multiple myeloma patients”. It is a well-designed and conducted study, showing authors previous experience in this field.

The authors thank the Reviewer for a thorough evaluation of the manuscript. We have carefully addressed the comments of all Reviewers. The changes introduced in the text of the manuscript have been marked using red font. Below, we present the detailed answers to the Reviewer’s comments and the description of the modifications introduced upon revision of the manuscript.

Here are my comments:

  1. The authors should try rewriting the Introduction section in order to avoid using multiple myeloma or MM so many time because, in the end, it can be annoying. It is aplicable to the Abstract section , as well.

We have corrected Abstract: “Background and Objectives: Anemia is common in multiple myeloma (MM) and is caused by complex pathomechanism including impaired iron homeostasis. Our aim was to evaluate biomarkers of iron turnover: serum soluble transferrin receptor (sTfR) and hepcidin-25 in patients at various stages of MM in relation with markers of anemia, iron status, inflammation, renal impairment and burden of this disease and as predictors of mortality. Materials and Methods: Seventy-three MM patients (six with smoldering type and 67 with symptomatic disease) were recruited and observed for up to 27 months. Control group included 21 healthy individuals. Serum sTfR and hepcidin were measured with immunoenzymatic assays. Results: MM patients with and without anemia had higher sTFR compared to controls, while only anemic examineted patients had higher hepcidin-25. Both hepcidin-25 and sTfR were higher in anemic than non-anemic patients. Higher hepcidin-25 (but not sTfR) was associated with increasing MM advancement (from smoldering type to International Staging System stage III disease) and with poor response to MM treatment, which was accompanied by lower blood hemoglobin and increased anisocytosis. Neither serum hepcidin-25 nor sTfR were correlated with markers of renal impairment. Hepcidin-25 predicted blood hemoglobin in MM patients independently of other predictors including markers of renal impairment, inflammation and burden of this disease. Moreover, both blood hemoglobin and serum hepcidin-25 were independently associated with 2-year patients’ survival. Conclusion: Our results suggest that hepcidin-25 is involved in anemia in MM and its concentrations are not affected by kidney impairment. Moreover, serum hepcidin-25 may be an early predictor of survival in this disease, independent of hemoglobin concentration. It should be further evaluated whether including hepcidin improves the early diagnosis of anemia in MM.” (lines 21-41).

And we have corrected Introduction: “Multiple myeloma (MM) is a plasma cell disorder that accounts for approximately 10% of hematological malignancies. It is more common disease in the elderly population and often develops following an asymptomatic stage. The diagnostic process involves the search for a myeloma defining event, including anemia and renal impairment. However, recent guidelines [1] allow for earlier diagnosis of MM preceding the development of organ damage in patients with high (≥60%) clonal plasma cell involvement of bone marrow, involved to uninvolved serum free light chains (FLCs) ratio ≥ 100 and serum FLCs ≥ 100 mg/L, or more than one focal lesion in magnetic resonance imaging. Median overall survival of patients with newly diagnosed disease reaches 6 to 8 years, which is a striking improvement attributed to modern therapy [2]. The focus on timely diagnosis and prevention of organ damage drives the interest in diagnostic and prognostic biomarkers in MM. In theory, a validated biomarker can act as a surrogate of underlying disease pathways and inform the physician on the alterations in organ-specific processes; for example, our previous studies showed that urinary insulin growth factor binding protein 7 (IGFBP-7), neutrophil gelatinase-associated lipocalin (NGAL) monomer and transgelin-2 may be markers of renal impairment in patients with MM [3,4]. The use of appropriate biomarkers may be crucial for adequate tailoring of the treatment plan. However, in order to be widely adopted into practice, novel biomarkers need to offer a significant improvement as compared to the current ones.

Anemia is commonly observed in MM (in about 70% of patients), and is even more prevalent in those patients who develop renal impairment (almost 90% of such cases).  In this disease, anemia may be caused by complex pathomechanisms. Cytokines produced by plasmacytes lead to anemia of chronic disease (ACD), by erythropoiesis inhibition and impaired iron homeostasis [5]. The most common laboratory findings in MM-related anemia and ACD are: (1) normocytic and normochromic anemia, (2) normal to mildly low serum iron levels, (3) high serum ferritin, (4) hemosiderin in bone marrow macrophages [6]. It seems that inadequate excretion of erythropoietin (EPO) compared to the degree of anemia, reduction of erythrocytes’ survival time (<10%), inadequacy of ferric management and direct suppression of erythropoiesis by neoplastic cells are mainly responsible for the development of ACD [7]. Other MM-related factors leading to anemia include: displacement of erythroid system by neoplastic plasmacytes, proinflammatory activity of cytokines and disabled apoptosis of the erythroid system. However, renal failure should also be taken into consideration as an important factor contributing to anemia in patients with this disease. Among MM patients whit renal impairment lower serum erythropoietin levels appear more frequently (up to 60%) than in patients with normal renal function [9]. The molecules engaged in iron metabolism, namely hepcidin and soluble transferrin receptor (sTfR), have been previously suggested as potential biomarkers in MM, enabling to better characterize the underlying etiology of anemia [9].

Hepcidin, as a one of acute phase proteins, is regulated by interleukin-6 (IL-6), a cytokine inducing MM development (a potential growth factor for myeloma cells). Hepcidin-25 (an active hormone consisting of 25 aminoacids) controls iron delivery from intestinal cells to the blood, regulates iron transport and its release from macrophages. There is a potential association between novel drugs in MM and this marker [9]. Moreover, hepcidin expression is regulated by growth differentiation factor 15 (GDF-15), secreted by tumor microenvironment cell in this disease. In our previous study, increased GDF-15 was associated with end-organ damage and adverse prognosis in MM patients [7].

sTfR prevents organism from iron-associated toxicity. TfR1 form has higher affinity for transferrin and is overexpressed on cells with a high rate of proliferation, including malignant hematopoietic cells. Interestingly, sTfR to ferritin ratio was decreased in advanced stages of hematopoietic malignancies Studies have shown increased sTfR serum levels, decreased ferritin and increased sTfR to ferritin ratio in correlation with increasing stages of chronic kidney disease (CKD) [10, 11].

The primary aim of this study was to evaluate biomarkers of iron turnover, sTfR and hepcidin-25 in patients at various stages of MM progression, and examine their relationship with indicators of anemia and iron status (hemoglobin, red blood cell indices, ferritin, serum iron concentration) and  renal impairment (serum creatinine, estimated glomerular filtration rate – eGFR and urinary NGAL) Moreover, we examined the relationships of these two markers with indicators of inflammation (IL-6) and burden of this disease (FLCs, β2-microglobulin, GDF-15). Finally, serum concentrations of sTfR and hepcidin-25 were assessed to verify them as predictors of mortality in multiple myeloma.” (lines 46-107).

  1. The primary and secondary objectives of this study should be better exposed.

Response: We have rephrased the aim of the study (lines 99-106): The primary aim of this study was to evaluate biomarkers of iron turnover, sTfR and hepcidin-25 in patients at various stages of MM progression, and examine their relationship with indicators of anemia and iron status (hemoglobin, red blood cell indices, ferritin, serum iron concentration) and  renal impairment (serum creatinine, estimated glomerular filtration rate – eGFR and urinary NGAL). Moreover, we examined the relationships of these two markers with indicators of inflammation (IL-6) and burden of this disease (FLCs, β2-microglobulin, GDF-15). Finally, serum concentrations of sTfR and hepcidin-25 were assessed to verify them as predictors of mortality in multiple myeloma.”

  1. The duration of the study should pe presented as a date, not only mentioning that it was about 27 months, in section “study design and patients”

Response: We have added date in Study Design and Patients section: “Follow-up data on mortality was collected in February 2020 after 27 months from the start of the study.” (lines 123-124).

  1. A flowchart showing the included and excluded patients can increase the value of the manuscript

Response: A flow diagram of patients selection was included as Figure 1.

  1. Lines 334-335: “”from among the studied iron…” – please rewrite

Response:  We have changed phrase: “Moreover, baseline serum hepcidin-25 appeared to be a negative predictor of survival, independent of haemoglobin concentration (Table 5).” (lines 334-335).

  1. Line 375 – “lone myeloma cells…” – what do you mean by that?

Response:  We have changed phrase: “Myeloma cells can promote erythroblast apoptosis [21,22], while cytokines, such as IL-6, impair erythroid maturation and hemoglobin production outside of “iron restriction” pathways [23].” (lines 374-375).

Reviewer 2 Report

Comments This is an interesting study, the authors collected samples from multiple myeloma and analyzed serum soluble transferrin receptor and hepcidin25. Overall, this paper is well-written and well-structured. However, in my opinion, there are some shortcomings in this paper regarding some data-taking methods and data analysis, which are not fully utilized.

  • As the author points out, there are many causes of anemia in MM patients. Treatment, of course, has an impact on anemia. There is a lot of variability in the patient background of MM patients. Whether the patient is pre-treatment or on treatment and the type of chemotherapy, will also affect the anemia. The authors should be corrected for in the analysis.
  • The authors analyzed the two-year overall survival using multiple cox proportion. How did you select variables? It is somewhat curious that hepcidin25 is a biomarker for survival, despite the fact that other promising biomarkers exist. What are the possible causes of the effect of hepcidin25 on survival?

Author Response

Manuscript ID: medicina-1618296

Manuscript title: The Key Role of Hepcidin-25 in Anemia in Multiple Myeloma Pa-Tients with Renal Impairment

Response to Reviewer 2

Comments This is an interesting study, the authors collected samples from multiple myeloma and analyzed serum soluble transferrin receptor and hepcidin25. Overall, this paper is well-written and well-structured. However, in my opinion, there are some shortcomings in this paper regarding some data-taking methods and data analysis, which are not fully utilized.

The authors thank the Reviewer for a thorough evaluation of the manuscript. We have carefully addressed the comments of all Reviewers. The changes introduced in the text of the manuscript have been marked using red font. Below, we present the detailed answers to the Reviewer’s comments and the description of the modifications introduced upon revision of the manuscript.

  1. As the author points out, there are many causes of anemia in MM patients. Treatment, of course, has an impact on anemia. There is a lot of variability in the patient background of MM patients. Whether the patient is pre-treatment or on treatment and the type of chemotherapy, will also affect the anemia. The authors should be corrected for in the analysis.

Response: In paragraph 3.2 (pages 7-8), we described the associations between the studied markers and the clinical characteristics of patients, including treatment (lines 273-285). Also, we included information about the association between blood hemoglobin and MM treatment (lines 278-283). The multiple regression analysis presented in Table 3, exploring the predictors of blood hemoglobin, has been adjusted for treatment status and the response to treatment (as stated in the Title of Table 3). However, we were not able to perform stratified Cox proportional hazard regression, because of limited number of studied patients. Please find a more detailed explanation of the Cox regression in the answer to your next comment.

  1. The authors analyzed the two-year overall survival using multiple cox proportion. How did you select variables? It is somewhat curious that hepcidin25 is a biomarker for survival, despite the fact that other promising biomarkers exist. What are the possible causes of the effect of hepcidin25 on survival?

Response: One of our aims was to verify whether the studied markers, i.e. serum hepcidin and soluble transferrin receptor (sTfR) are associated with patients’ survival. Therefore, we first used the studied markers as the predictor variables in simple Cox regression (i.e. hepcidin-25 and sTfR were assessed separately). Thereafter, as we observed that serum hepcidin was associated with survival in simple analysis, we checked if this association is independent from hemoglobin concentration (also a significant predictor of survival in simple analysis). The missing hazard ratios calculated in simple analysis have been added upon revision (lines 143-148). The limitation of our study is a relatively small group of participants; for that reason, we were not able to perform stratified Cox regression including the treatment status and the response to treatment. We rephrased our conclusions, highlighting that the observed association between serum hepcidin concentrations and survival must be considered preliminary due to the limitations of our study, and should be verified in larger studies (lines 428-430 in Discussion, line 483 in Conclusions).

We have added comment in Discusion section: “Hepcidin levels are increased in many malignancies including MM, which provides essential source of iron for their survival. Neoplasms cells require increasing cellular iron import by actions of hepcidin on ferroportin downregulation. In MM hepcidin expression is related to bone morphogenetic protein 2 (BMP2) and IL-6. However, BMP2 is stronger inducer of hepcidin than IL-6 [36]. Moreover, local levels of hepcidin are lower than serum levels. All of that suggest that hepcidin mainly derives from the liver in MM [37] It explains our observation of no correlation between hepcidin and IL-6.” (lines 414-421).

Round 2

Reviewer 2 Report

The authors carefully answered the reviewer's comments. They also mentioned the limitation of this study.